# Modification Effects of Population Expansion, Ageing, and Adaptation on Heat-Related Mortality Risks Under Different Climate Change Scenarios in Guangzhou, China

**DOI:** 10.3390/ijerph16030376

**Published:** 2019-01-29

**Authors:** Tao Liu, Zhoupeng Ren, Yonghui Zhang, Baixiang Feng, Hualiang Lin, Jianpeng Xiao, Weilin Zeng, Xing Li, Zhihao Li, Shannon Rutherford, Yanjun Xu, Shao Lin, Philip C. Nasca, Yaodong Du, Jinfeng Wang, Cunrui Huang, Peng Jia, Wenjun Ma

**Affiliations:** 1Guangdong Provincial Institute of Public Health, Guangdong Provincial Center for Disease Control and Prevention, Guangzhou 511430, China; gztt_2002@163.com (T.L.); fengbaixiang@126.com (B.F.); jpengx@163.com (J.X.); zengwl@gdiph.org.cn (W.Z.); lixing.echo@foxmail.com (X.L.); zhihaoli1990@163.com (Z.L.); 2State Key Laboratory of Resources and Environmental Information System, Institute of Geographic Sciences and Natural Resources Research, Chinese Academy of Sciences, Beijing 100101, China; renzp@Lreis.ac.cn (Z.R.); wangjf@Lreis.ac.cn (J.W.); 3Guangdong Provincial Center for Disease Control and Prevention, Guangzhou 511430, China; zyh@cdcp.org.cn (Y.Z.); gdxyj05@21cn.com (Y.X.); 4School of Public Health, Sun Yat-sen University, Guangzhou 510080, China; hualianglin@gmail.com (H.L.); huangcr@mail.sysu.edu.cn (C.H.); 5School of Medicine, Griffith University, Brisbane QLD 4111, Australia; s.rutherford@griffith.edu.au; 6Department of Environmental Health Sciences, School of Public Health, University at Albany, State University of New York, One University Pl, Rensselaer, NY 12148, USA; slin@albany.edu (S.L.); pnasca@albany.edu (P.C.N.); 7Guangdong Provincial Climate Center, Guangzhou 510080, China; yddu@grmc.gov.cn; 8Department of Earth Observation Science, Faculty of Geo-information Science and Earth Observation (ITC), University of Twente, 7500 Enschede, The Netherlands; jiapengff@hotmail.com; 9International Initiative on Spatial Lifecourse Epidemiology (ISLE), 7500 Enschede, The Netherlands

**Keywords:** climate change, years of life lost, population expansion, ageing, adaptation, population health

## Abstract

(1) Background: Although the health effects of future climate change have been examined in previous studies, few have considered additive impacts of population expansion, ageing, and adaptation. We aimed to quantify the future heat-related years of life lost (*YLLs*) under different Representative Concentration Pathways (RCP) scenarios and global-scale General Circulation Models (GCMs), and further to examine relative contributions of population expansion, ageing, and adaptation on these projections. (2) Methods: We used downscaled and bias-corrected projections of daily temperature from 27 GCMs under RCP2.6, 4.5, and 8.5 scenarios to quantify the potential annual heat-related YLLs in Guangzhou, China in the 2030s, 2060s, and 2090s, compared to those in the 1980s as a baseline. We also explored the modification effects of a range of population expansion, ageing, and adaptation scenarios on the heat-related *YLLs*. (3) Results: Global warming, particularly under the RCP8.5 scenario, would lead to a substantial increase in the heat-related *YLLs* in the 2030s, 2060s, and 2090s for the majority of the GCMs. For the total population, the annual heat-related *YLLs* under the RCP8.5 in the 2030s, 2060s, and 2090s were 2.2, 7.0, and 11.4 thousand, respectively. The heat effects would be significantly exacerbated by rapid population expansion and ageing. However, substantial heat-related *YLLs* could be counteracted by the increased adaptation (75% for the total population and 20% for the elderly). (4) Conclusions: The rapid population expansion and ageing coinciding with climate change may present an important health challenge in China, which, however, could be partially counteracted by the increased adaptation of individuals.

## 1. Introduction

The Intergovernmental Panel on Climate Change (IPCC) has projected that the increase in global surface temperature will continue in the coming decades [1]. The impacts of high temperature on human health have been reported in many previous studies [2,3]. Hence, the health risk assessment of future climate change could aid in improving the design of public health interventions and policies, preparedness of adaptation strategies, and healthcare planning. Numerous studies have assessed the heat-related mortality risks of future climate change in developed countries, but few were performed in developing countries where health vulnerability to climate change is greater [4,5]. In addition, in the process of assessing health risks, few studies have considered the factors affecting the vulnerability and susceptibility of humans to increasing temperature, such as population ageing and adaptation [6,7,8,9,10]. 

Ageing is an important determinant of human vulnerability to increasing temperature, as the elderly are more at risk from extreme heat events [11,12]. In addition, people can also acclimatize to climate change through physiological and technical adaptations [13,14,15]. Therefore, it is vital to integrate these factors into the health risk assessment of future climate change, which can broaden our understanding of the emerging health risks caused by climate change.

In this study, we projected the future heat-related years of life lost (*YLLs*) in Guangzhou, China, for the 2030s, 2060s, and 2090s under three Representative Concentration Pathways (RCP) scenarios (RCP2.6, 4.5, and 8.5) and 27 global-scale General Circulation Models (GCMs). The relative contributions of population expansion, ageing, and adaptation to these heat-related *YLLs* were also examined.

## 2. Materials and Methods 

### 2.1. Study Settings

Guangzhou city is the third largest metropolis and the capital of Guangdong Province in South China (Appendix A). In 2010 it had a population of 12.7 million, of which 6.62% were 65 years and over [16]. The ambient temperature in Guangzhou increased at a rate of 0.13 °C per decade during 1951–2004, which was higher than the national average rate (0.04 °C per decade). In particular, the ambient temperature increased more rapidly after the 1980s, which makes it one of the cities with the rapidest increase in temperature in China [17]. In addition, it is expected that the ambient temperature in Guangzhou will continue to increase in future decades [1].

### 2.2. Data Collection and Preparation

Daily non-accidental mortality data in Guangzhou during 2010–2015 were obtained from Guangdong Provincial Center for Disease Control and Prevention (GDCDC). Non-accidental deaths were categorized using A00-R99 codes from the International Classification of Diseases 10th Revision (ICD-10). *YLLs* were calculated by matching each death by age and sex to the life table of China for the year 2010 [18]. Total daily *YLLs* were calculated by summing the *YLLs* for all deaths on the same day. We also estimated the daily *YLLs* for males, females, and people aged <65 and ≥65 years, respectively.

Historical daily meteorological data, including daily mean temperature (*TM*), relative humidity (*RH*) and wind speed (*WS*), for the 1980s and for 2010–2015 in Guangzhou, were collected from the Guangdong Provincial Meteorological Bureau. The 1980s were chosen as our modeling baseline because this decade is at the center of the conventional climatological baseline period from 1971 to 2000, which was also employed in several previous studies [19,20]. The *YLLs* estimated during the baseline period were subtracted from heat-related *YLLs* for future periods. Meteorological data during the 2010–2015 were used to assess the exposure-response relationship between high temperature and YLLs in Guangzhou, because these data are the most updated and hence can be matched to the most recent census data in 2010.

Future daily temperature data projected using 27 GCMs under three RCP scenarios were collected from Coupled Model Inter-comparison Project 5 (CMIP5) (Appendix A) [1,21]. A single ensemble was used for each GCM model. We chose RCP2.6, RCP4.5, and RCP8.5 scenarios to represent the low, middle, and high greenhouse gas (GHG) emissions, respectively. Because GCMs often provide biased simulated temperature which are usually at coarse spatial resolution [22], we employed three bias-correction models (*unbiasing*, *qqmap*, and *isi-mip*) to downscale the projected temperature data to a finer spatial resolution of 0.5 °C × 0.5 °C [23], in which the daily TM data of 680 meteorological stations during 1960–1999 in China [24] was used as a training dataset. We finally selected the results derived from *isi-mip* as input for projection as it outperformed the other two methods (Appendix A). The output corresponding to the geographical location of Guangzhou was used.

Daily air pollutant data, including NO_2_ (nitrogen dioxide), SO_2_ (sulfur dioxide), and PM_10_ (particulate matter with an aerodynamic diameter of 10 μm or less) during 2010–2015 were obtained from Guangzhou Environmental Monitoring Center.

Population expansion was indicated by the total population size. The population information (total population size and its sex and age structures) in Guangzhou in the 2030s, 2060s and 2090s was projected using the framework employed by the United Nations Population Estimates and Projections [25,26]. Based on this framework, we integrated historical sex- and age-specific mortality rates, total fertility rate (*TFR*), population counts (population size and migration), life expectancy at birth and sex ratio from 5-year periods during 1950–2015 in Guangzhou. The historical (1950–2015) and projected *TFRs* (2020–2100), and historical period sex- and age-specific mortality rates (1950–2015) were obtained from the United Nations provided at the Chinese national level data, as the Guangzhou city specific data were not available. However, available mortality data for several years (i.e., 2010) indicated that the mortality rates for the total population, males, and females in Guangzhou were similar to the national levels [27]. The historical population size and sex ratio data were collected from the Guangzhou Statistical Yearbook [28]. Historical life expectancy at birth were obtained from several previous studies [29,30,31]. Based on these historical data, we first generated the probabilistic projections of sex-specific life expectancies at birth for 2020–2100 using a Bayes hierarchical model [32]. A sample of 10,000 trajectories of future life expectancy were estimated to show their probability distribution. Then the period probabilistic population projections for 2020–2100 were estimated using the standard cohort-component model [33]. This method provides a sample of 10,000 values of any future population quality to approximate its predictive distribution. To consider the uncertainty of projection, we employed the median and 90% interval of projection trajectories to respectively represent the medium, low and high scenarios of population index increases in the future: total population size, sex-and age-specific population size, and proportion of the elderly population (≥65 years) (Please refer to Appendix A for details).

Adaptation was defined by the IPCC as “the process of adjustment to actual or expected climate and its effects” [34]. The main types of climate change adaptation include physiological adaptation (or acclimatization), behavioral (e.g., air conditioning), infrastructure (e.g., healthcare systems), and technological adaptation (e.g., heat warning systems) [15]. In this study, we employed three adaptation scenarios in this study to capture their effects on the heat-related health effects. In the first scenario (S1), we assumed that people’s adaptation would increase by 8.92% per decade [35], which was estimated by Yang’s study conducted in Shanghai that has the similar socioeconomic characteristics with Guangzhou. In Yang’s study, they assessed the long-term variation in the association between ambient temperature and daily cardiovascular mortality in Shanghai from 1981 to 2012. Their findings showed that the extremely hot effects (99th percentile of mean temperature) decreased by 8.92% per decade. This is a unique study that has assessed the long-term variation of heat effects in China. However, the 30-year study duration might not totally capture the long-term variation of people’s adaptation. In addition, Yang et al. assessed only heat effects on cardiovascular mortality. Therefore, we used another adaptation scenario (S2) that was from Petkova et al.’s study conducted in New York City. This study quantitatively assessed the effects of people’s adaptation on daily temperature impacts over a period spanning more than a century [36]. They observed that a decrease in heat effects of 4.6% per decade could be attributed to people’s adaptation. In addition, people’s adaptation to high temperature could also be measured by the change of minimum mortality temperature (*MMT*) that is the temperature with the lowest mortality risk. The increase of *MMT* over time suggests improved adaptation to heat effects [37]. In a recent study, Todd and Valleron assessed the temperature–mortality relationship in France from 1968 to 2009, and observed that the *MMT* increased by about 0.2 °C/decade [38]. In this study, we also employed the change of *MMT* to assess people’s adaptation (S3), and assumed (consistent with Todd and Valleron, 2015) that the *MMT* would also increase by 0.2 °C per decade in Guangzhou.

### 2.3. Heat Effects Estimation

We first employed a distributed lag non-linear model (DLNM) [39] to estimate the non-linear and lag effects of heat effects on *YLLs* for all deaths during 2010–2015. The model can be written as:*YLL_t_* = *α* + *βT_t,l_*(TM) + *ns*(RH, *df*) + *ns*(WS, *df*) + *ns*(time*_t_*, *df*) + *ns*(SO_2_, *df*) + *ns*(NO_2_, *df*) + *ns*(PM_10_, *df*) + *η*DOW(1)
where *t* denotes the day of observation; *YLL_t_* denotes the total *YLLs* on day *t*; *α* denotes the intercept indicting the baseline risk. *T_t,l_* is a matrix obtained by applying the DLNM to *TM*; *β* denotes the vector of coefficients for *T_t,l,_* and *l* denotes the number of lag days. We employed a B-spline function (*bs*) and a natural cubic spline function (*ns*) to estimate the non-linear and lagged effect of *TM*, respectively. We fitted a lag structure of up to 1 day (lag 0–1) according to our preliminary analysis which showed that the heat effects mainly appeared during the first two days (Appendix A). The family function for DLNM was Gaussian. Degrees of freedom (*df*) for the lag structure were chosen based on Akaike information criterion (AIC) [40]. It was found that 3*df*s for non-linear effects of *TM* produced the best model fit. The *df*s for RH, WS, SO_2_, NO_2_, and PM_10_ were all set to 3, consistent with some previous studies [41,42]. 5*df*s per year was used to control for secular trend indicated by *time* that equals 1, 2, 3, … 2191 (day of the study period 2010–2015). DOW is a dummy variable representing day of the week, and *η* is a vector of coefficients.

Based on equation (1), a healthy temperature range (*TM*) of 12.6–21.0 °C, within which the heat effects were not statistically significant, was identified. A temperature of 21.0 °C was defined as the threshold temperature, and all the cumulative effects (*YLLs*) of temperature above 21.0 °C along lag 0–1 days were defined as heat effects. Similarly, we defined the heat effect threshold temperature in males (24.3 °C), females (21.5 °C), and people <65 (28.5 °C) and ≥65 years (elderly) (18.6 °C).

### 2.4. Projection of Future Heat Effects

The heat-related *YLLs* under each RCP scenario (RCP2.6, 4.5 and 8.5) and GCM were estimated using modelled daily *TM* in the 1980s, 2030s, 2060s, and 2090s, respectively. The calculation process is described as:(2)$$YLL^{\prime }=\overset{n}{\underset{t=21.0}{\sum }}YLL_{t}\times N_{TM}$$
where *YLL*′ denotes the total heat-related *YLLs*. *YLL_t_* denotes the attributable *YLLs* of *TM* ≥ 21.0 °C which was obtained by Equation (1); *n* is the maximum of daily TMs during each study period. *N_TM_* is the average annual number of days with TM ≥ 21.0 °C. Then, we estimated the differences in annual heat-related YLLs between the future and 1980s as a baseline under each RCP scenario and each GCM. Here, we assumed that the population size and their adaptation in the 2030s, 2060s and 2090s would remain constant at the 2010 level.

To test the independent impacts of population expansion on the future heat-related *YLLs*, we estimated the population expansion adjusted heat-related *YLLs*:(3)$$YLL_{p}^{\prime }=YLL^{\prime }\times U_{20X0}/U_{2010}$$
where *YLL′_p_* denotes the annual heat-related *YLLs* after taking into account the population expansion level. *U_20X0_* denotes the projected population expansion in Guangzhou in the future (2030s, 2060s and 2090s), and *U_2010_* denotes the population size in Guangzhou in 2010. Similarly, we calculated the population expansion adjusted annual heat-related *YLLs* (*YLL′_ep_*) in the elderly.

While assessing the impacts of the degree of ageing (percentage of elderly ≥65 years in the total population) on the future heat-related YLLs, the impacts of total population size could not be ignored, because more people will enter this age range along with the increase in population size. Therefore, we used Equation (4) to estimate the heat-related YLLs for the elderly in every one million total population, and then explored the relationship between the degree of ageing and the heat-related *YLLs* for the elderly, which could be used to adjust for the bias of total population sizes:(4)$${{YLL}^{\prime }}_{p/m}={{YLL}^{\prime }}_{ep}/P_{20X0}$$
where *YLL′_p/m_* denotes the annual heat-related *YLLs* of the elderly for every one million total population under different scenarios of population increase. *YLL′_ep_* denotes the annual heat-related *YLLs* for the elderly in the future after taking into account the population expansion. *P_20X0_* denotes the total population size (million) in Guangzhou in the future.

We also assessed the impacts of adaptation on the heat effects in the future. We considered three adaptation scenarios. In adaptation S1, people’s adaptation was assumed to increase by 8.92% per decade [35]:(5)$$YLL_{a}^{\prime }=YLL^{\prime }\times {\left(1-0.0892\right)}^{Z}$$
where *YLL′_a_* denotes the annual heated-related *YLLs* in the future after deducting the heat effects offset by people’s increasing adaptation. 0.0892 (8.92%) is the decrease in rate of heat effects per decade. *Z* denotes the difference in number of decades between future study projection years (2030s, 2060s, and 2090s) and 2010. Similarly, we estimated the adjusted heated-related *YLLs* in the future if people’s adaptation increased by 4.60% per decade [36]. In adaptation S3, we assumed that the MMT in the nonlinear relationship between TM and YLLs would increase by 0.2 °C/decade [38]. For example, the (*TM* + 0.4) in the 2030s would lead to the same heat-related health effects with *TM* in the 2010s. Equation (5) is used to illustrate the process in the total population as an example: (6)$$YLL_{a3}^{\prime }=\overset{n}{\underset{t=21.0}{\sum }}YLL_{t}\times N_{TM-Z\times 0.2}$$
where $YLL_{a3}^{\prime }$ denotes the annual heated-related *YLLs* in the future. *TM* denotes the daily mean temperature in the future. *N*_*TM* − Z × 0.2_ denotes the annual number of days with (*TM* − *Z* × 0.22) ≥ 21.0 °C. *Z* denotes the difference in number of decades between future study projection years (2030s, 2060s and 2090s) and 2010. *YLL_t_* is the attributable *YLLs* on day *t* with a (*TM* − *Z* × 0.2) ≥ 21.0 °C as compared with the *YLLs* in days with the healthy temperature. *n* is the maximum daily temperature (*TM* − *Z* × 0.2). For example, the attributable *YLLs* on the day with a TM of 25.2 °C in the 2030s was defined as the average *YLLs* on the day with a TM of 25.0 (25.2−0.4) °C in the 2010s in Guangzhou.

Finally, we estimated the impacts of both adaptation and population expansion on the heat effects: (7)$$YLL_{pa}^{\prime }=YLL^{\prime }\times \frac{U_{20X0}}{U_{2010}}\times {\left(1-0.0892\right)}^{Z}$$
where *YLL′_pa_* denotes the annual heated-related *YLLs* in the future after taking into account both population expansion and adaptation; *U_20X0_* and *U_2010_* denote the population expansion levels in the future and 2010s respectively. Similarly, we also estimated the *YLL′_pa_* under the other two scenarios of adaptation.

All above calculation processes were conducted for males, females, and people aged <65 and ≥65 years. We used R software (version 3.4.0; R Development Core Team 2012, http://www.R-project.org/, Vienna, Austria) to fit all models.

## 3. Results

### 3.1. General Characteristics

In Guangzhou during 2010–2015, the annual *TM* was 21.9 °C and the average total daily *YLLs* was 2408.7 (Table 1). The 27GCM outputs indicate that temperature will increase more rapidly under the RCP8.5 and RCP4.5 scenarios than the RCP2.6 scenario, and the average increase in temperature was 0.023, 0.125, and 0.320 °C per decade for the three scenarios, respectively (Figure 1 and Appendix A). According to the population projections, the total population in Guangzhou will peak during the 2030s and then decrease after 2040 under the low and medium scenarios, but it will keep increasing in the high scenario (Appendix A). We observed a typical U-shaped relationship between *TM* and *YLLs* along lag 0−1 days from 2010 to 2015 (Figure 2 and Appendix A).

### 3.2. Independent Effects of Temperature Increase on Heat-Related Ylls in the 2030s, 2060s, and 2090s

Without integrating urbanization and adaptation, we observed a predominant increasing trend in the *YLLs* for the three future periods under all RCP scenarios and the majority of 27 GCMs, although there were large variations in *YLLs* among GCMs. The increments of *YLLs* were more rapid under the RCP4.5 and RCP8.5 scenarios compared with the RCP2.6 scenario. For the total population, the annual heat-related YLLs in the 2030s, 2060s, and 2090s were 1.6, 2.4, and 2.5 thousand, respectively, under RCP2.6, and the corresponding figures were 2.2, 7.0, and 11.4 thousand, respectively, under RCP8.5 compared with the baseline range (Figure 3 and Appendix A).

### 3.3. Modification of Population Expansion and Adaptation on Heat-Related YLLs in the 2030s, 2060s, and 2090s

During the 2030s, when the population will reach its peak, comparing with the scenario of constant population size at the 2010 level, we observed a more rapid increase in heat-related YLLs from low, medium to high population expansion under all climate change scenarios if the adaptation was constant. Each 10% increase in population expansion level was associated with an average of 4.23, 4.23, and 4.29 thousand more *YLLs* under the RCP2.6, RCP4.5, and RCP8.5 scenarios, respectively, in the 2030s (Appendix A). For the 2060s, we observed a similar trend in heat-related *YLLs* as the 2030s from low, medium to high expansion scenarios, but the *YLLs* under the low expansion scenario were lower than the constant expansion scenario. In terms of the 2090s, *YLLs* under low as well as medium expansion scenarios are smaller than the constant expansion scenario, but the changing trend in *YLLs* from low, medium to high expansion is consistent with the 2030s and 2060s. One of the major reasons for the *YLL* decrease is a decrease in population size under the low and medium scenarios.

After considering adaptation, we found that heat-related *YLLs* would be largely counteracted under adaptation S1 (8.92% decrease per decade) and S3 (0.2 °C increase in MMT per decade) scenarios, but less modified under adaptation S2 scenario (4.60% decrease per decade) for the total population. For example, 78.0%, 77.9%, and 74.4% of heat-related *YLLs* under high population expansion scenarios in the 2030s could be offset by adaptation (S1) under the RCP2.6, RCP4.5, and RCP8.5 scenarios, respectively, and the figures declined to 14.7%, 14.8%, and 14.1%, respectively, under the adaptation S2 scenario. Furthermore, the heat-related YLLs could also be subsequently offset by adaptation in the 2060s and 2090s (Figure 4 and Appendix A).

We found future changing patterns of heat-related *YLLs* in the total population to be similar to patterns in males, females and the elderly population. In particular, we observed more rapid increases of heat-related *YLLs* in the elderly due to the rapid ageing of the population. Detailed information is shown in Appendix A.

### 3.4. Modification of Population Ageing and Adaptation on Heat-Related YLLs in the 2030s, 2060s, and 2090s

Figure 5 shows that the increased ageing (as measured by percentage of the elderly in the total population) will result in a significant increase in heat-related *YLLs* when adaptation is considered to be constant since 2010. Each 1% increase in the degree of ageing will lead to an average of 428.0 more *YLLs* in one million total population in the 2030s. Moreover, the ageing induced heat-related *YLLs* will consecutively increase as time progresses, and reach a peak in the 2090s. The heat-related *YLLs* in the future could be largely counteracted under the S1 and S3 adaptation scenarios, particularly in the 2060s and 2090s when people’s adaptation consecutively increases. The average *YLLs* for each 1% increase in ageing decreased from 428.0 to 368.5. The offset effects of the S2 adaptation were less than S1 and S3 (Figure 5).

## 4. Discussion

Climate change is not only a significant environmental issue, but also a serious public health challenge that may continue to grow as the planetary temperature increases. In this study, we found that heat-related *YLLs* in Guangzhou, China would dramatically increase under all climate change scenarios and most GCMs as the 21st century progresses. This finding was consistent with those of some previous studies. For example, in comparison with the 1980s, Li et al. identified that cardiovascular mortality in Beijing under the RCP8.5 scenario would increase by 16.6%, 73.8%, and 134.0% in the 2020s, 2050s, and 2080s, respectively [6]. These findings highlighted the potential public health benefits that could result from the scenario of lower GHG emission, and the critical importance of adequately adapting to climate change to protect vulnerable populations if we cannot reverse the trend of global warming in the near future.

We further found that rapid population expansion would significantly aggravate adverse heat-related effects under all climate change scenarios. It has been demonstrated that rapid population expansion, especially in urban areas, could lead to land use/cover change, pollution, overcrowding, changes in physical activity patterns, and inadequate service capacity for sanitation. All of them could elevate the human risk for climate change [8]. Hence, it is possible that urban areas where the population has been growing fastest are also those that are least equipped to deal with the threat of climate change.

We employed three population expansion scenarios (i.e., low, medium, and high) to assess uncertainties in the influences of future population expansion on heat effects, and observed that heat-related *YLLs* would decrease under the low and medium expansion scenarios in the 2060s and 2090s. One major reason for the reduction in *YLLs* is a decrease in population size under the low and medium scenarios, where the population would consecutively decline after 2040. However, the population expansion process in Guangzhou would probably follow the high scenario because of the universal two-child policy issued by the Chinese government in October 2015, which aimed to attenuate negative effects of the one-child policy, such as accelerating population ageing, the skewed sex ratio, and the decline of working-age populations [43]. It has been projected that the total fertility rate in urban areas would significantly increase in the next several decades [44], which will lead to a rapid increase in population size in megacities such as Guangzhou. These results suggest that the rapid population expansion will lead to unprecedented challenges in dealing with health threats of future climate change, and the heat-related health risks would be greatly underestimated if population expansion was not integrated into assessment models. Policy makers need to adequately consider actions to enhance sustainable development and urban resilience to the health impacts of climate change, such as through the reduction in GHG emissions (mitigation) and the increase in adaptation [45]. 

It has been observed that the elderly were more vulnerable to climate change [11,12] Ageing can decrease an individual’s tolerance to heat. The elderly also often suffer from comorbidity, social isolation, physical and cognitive impairment, and the need to take multiple medications, which may further increase their vulnerability to extreme heat [46]. Here, we found that the heat-related *YLLs* in the elderly were projected to increase more rapidly than other age groups. For example, we estimated that every 1% increase in the percentage of the elderly per one million population would be associated with an increase in around 420 heat-related *YLLs* in the 2030s; this figure would increase further in the 2060s and 2090s. A recent study conducted in Beijing observed similar results [6]. In the coming decades, China will go through a rapidly ageing stage due to the baby boom several decades ago. We estimated that the number of the elderly under the medium scenario in Guangzhou would increase from 0.84 million (6.62%) in 2010 to 3.5 (23.7%), 4.9 (35.7%), and 4.2 million (36.0%) in the 2030s, 2060s, and 2090s, respectively. Although the implementation of the universal two-child policy may partially alleviate the degree of ageing in the future, the absolute number of the elderly would be very large in the future [43]. Hence, the protection for the elderly should be prioritized in dealing with the adverse effects of global warming. Strategies that have enhanced the care of the elderly and improve their ability to cope with climate change, such as regular monitoring of their health conditions, encouraging suitable clothing, providing cool environments, appropriate diets, and adequate intake of fluids, need to be implemented particularly at the community level as most elderly people live with their families in the community.

Our results further revealed that adaptation would significantly offset the future heat effects, which was consistent with previous studies’ findings. For instance, Jenkins et al. observed that increasing the temperature threshold by 1 °C could result in declines in annual heat-related mortality ranging from 32% to 42% across the scenarios in London in the 2050s [47]. These findings repeatedly reminded us of the vital importance of adaptation in response to climate change, and that strategies such as heat plans [48] and other specific measures should be implemented to improve people’s adaptation. For example, properly designed infrastructure can provide physical protection; well-designed communications and early warning systems can help people to respond and cope with heat waves; and appropriate urban planning including land-use changes, building regulations, external shading, green space increase, and GHG mitigation can reduce heat exposure and hence help to reduce heat impacts.

Projecting future adaptation to heat is an important challenge in assessing the heat-related mortality of climate change. In previous studies, methods used to account for adaptation included analogue cities, analogue summers, and assuming adaptation to heat for a pre-determined number of degree Celsius [49], but very few used the past declines in vulnerability to temperature [6,7]. In addition, the adaptation may vary along with the process of climate change. For example, more intensive warming process may enhance people’s adaptive capacity. Therefore, we employed three adaptation scenarios in this study to assess uncertainties of offset effects of adaptation on heat-related *YLL* estimations. Of the three scenarios, two were assessed based on the past declines in vulnerability to temperature, and one was based on pre-determined change in temperature units of degrees Celsius. We observed that the adaptation S1 scenario would bring larger offset effects than the other two scenarios. The adaptation S1 scenario might be the most accurate trajectory for people in Guangzhou in the future. We believe that these estimates of heat effects derived from a broad range of scenarios provides uncertainty information for policy makers concerned with adaptation planning for climate change and health. 

### Limitations and Uncertainties

Some other limitations and uncertainties should be acknowledged in this study. The first source comes from the projection of population expansion. Although we have thought over the uncertainty, and employed a probabilistic model in our projection, we ignored the variation of vulnerability to temperature between immigrants from other places and the residents in Guangzhou because we could not project the age and gender structures of these immigrants. However, rural residents in China are more vulnerable to high temperature than urban residents, due to such factors as greater age, lower education, lower incomes, and less access to high quality medical care. Second, we did not adjust for other air pollutants such as O_3_, CO, and PM_2__.5_ in the DLNM because of lack of data, which may have led to potentially biased exposure-responses between temperature and *YLLs*. However, previous studies have demonstrated that the effects of air pollutants would be smaller than temperature, and did not significantly alter the effect sizes of temperatures [50]. Furthermore, some air pollutants such as O_3_ might be on the causal path between temperature and health, which can also complicate interpretation of adjusted models. Thirdly, we observed weaker associations of TM with YLLs in males and younger people. For example, the YLLs were significantly increased for people <65 years when TMs were larger than 28.5 °C, which may lead to underestimation of the heat effects. Another source of uncertainty is that we did not address the cold-related YLLs, although the reduction of cold-related mortality due to the warming climate change may partially offset the heat-related YLLs. Uncertainties may also come from some other sources, such as possible changes in humidity, air quality affected by climatic factors, health conditions of people, profile of the heatwave, intraurban variation of temperatures and risks, and changing of land-cover, building environments, quality and type of building stock, and population density. These uncertainties are expected to be included in future studies, and thus moving towards a more comprehensive risk-type framework that explicitly represents uncertainty in the assessment.

## 5. Conclusions

Our study comprehensively quantified the heat-related *YLLs* of future warming climate under the RCP scenarios on a local scale in China, observing an escalation of heat-related *YLLs* under all scenarios in the 21st century. The heat effects may increase dramatically with continued rapid population expansion processes and population ageing in the future. However, the adaptation increase would partially offset the heat-health effects. Therefore, it is necessary to design and implement more effective response strategies and measures in the future to reduce the heat risk in highly populated urban areas, by reducing heat exposure in all populations, improving their adaptation, and targeting most efforts on the elderly. 

## Figures and Tables

**Figure 1 ijerph-16-00376-f001:**
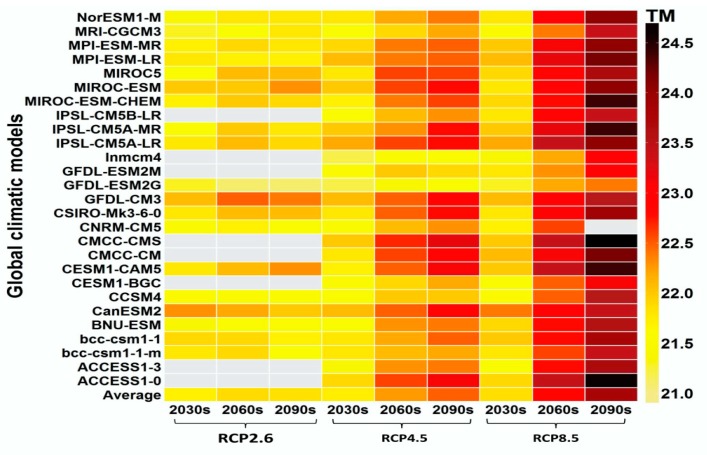
Annual temperature in the 2030s, 2060s, and 2090s under different climatic models and RCP scenarios. Gray grids mean that the data were not available.

**Figure 2 ijerph-16-00376-f002:**
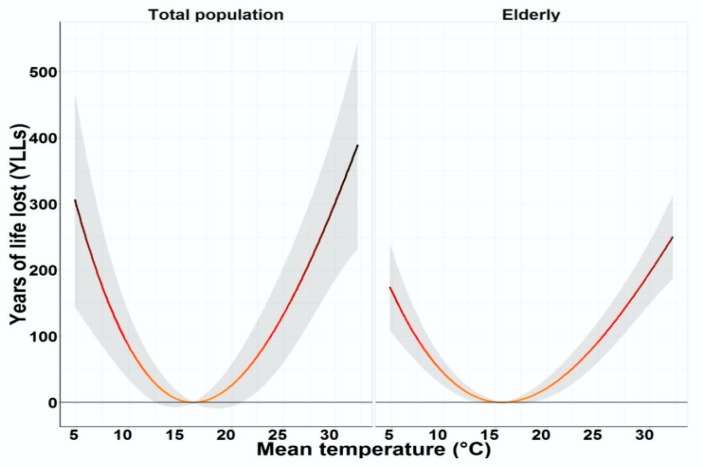
Relationships between daily mean temperature and *YLLs* in the total population and the elderly. All the effects of temperature on *YLLs* were adjusted for secular trend, wind speed, day of week, relative humidity, SO_2_, NO_2_, and PM_10_.

**Figure 3 ijerph-16-00376-f003:**
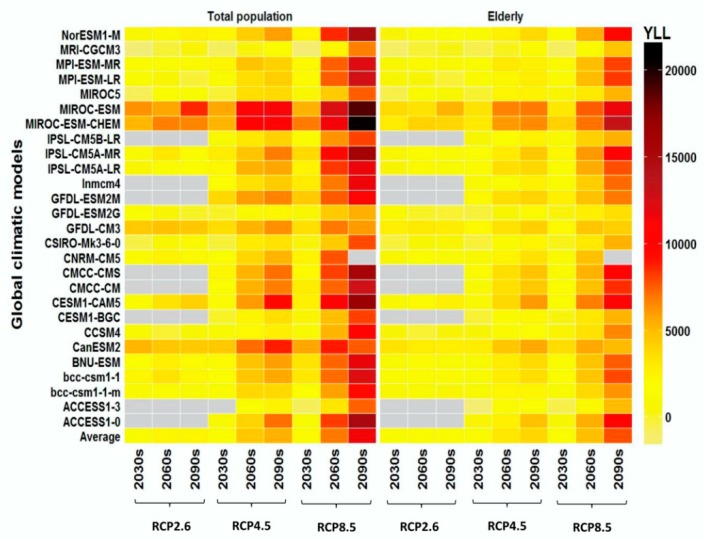
Annual heat-related *YLLs* in the total population and the elderly under different climatic scenarios. Note: The heat-related *YLLs* in the future have been subtracted by the heat-related *YLLs* in the 1980s. We assumed that the population size and their adaptation in the 21st century will remain constant at the 2010 level. Gray grids mean that the data were not available.

**Figure 4 ijerph-16-00376-f004:**
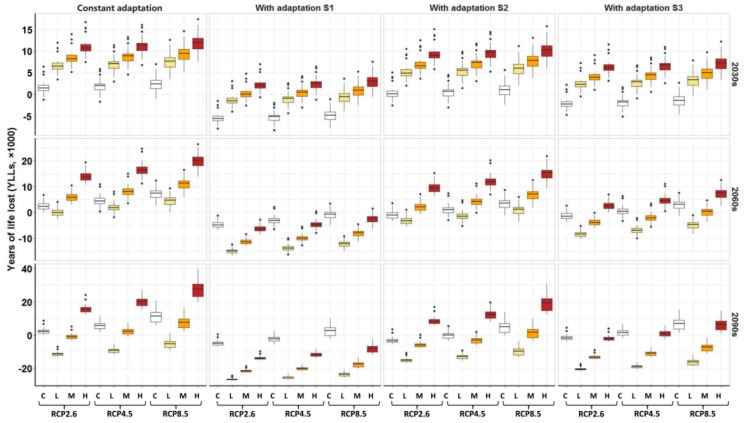
Impacts of population expansion and adaptation on the annual heat-related *YLLs* in the future. Constant adaptation: People’s adaptation to high temperature will remain constant at the 2010 level. Adaptation S1: People’s adaptation to high temperature will increase by 8.92% per decade. Adaptation S2: People’s adaptation to high temperature will increase by 4.60% per decade. Adaptation S3: People’s adaptation to high temperature will increase by 0.2 °C per decade. C: The population size will remain constant at the 2010 level. L: Low population expansion scenario. M: Medium population expansion scenario. H: High population expansion scenario. The three rows of panel show the effects of population expansion and adaptation on the heat-related *YLLs* in 2030s, 2060s, and 2090s, respectively. The heat-related *YLLs* in the future have been subtracted by the heat-related *YLLs* in the 1980s.

**Figure 5 ijerph-16-00376-f005:**
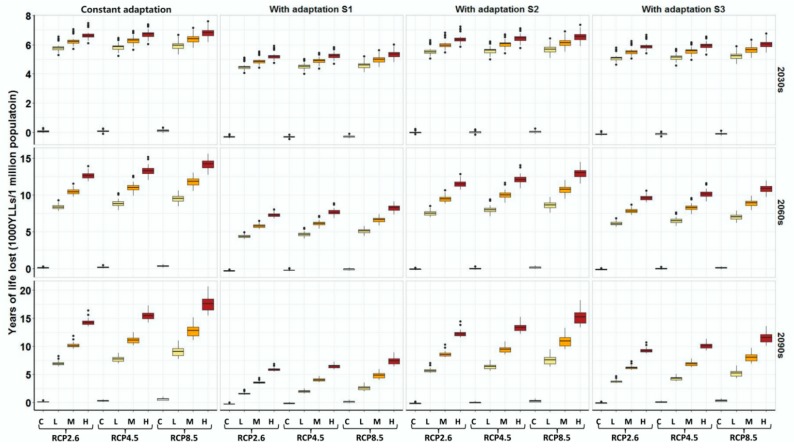
Impacts of aging and adaptation on the annual heat-related *YLLs* in the future. Constant adaptation: People’s adaptation to high temperature will remain constant at the 2010 level. Adaptation S1: People’s adaptation to high temperature will increase by 8.92% per decade. Adaptation S2: People’s adaptation to high temperature will increase by 4.60% per decade. Adaptation S3: People’s adaptation to high temperature will increase by 0.1 °C per decade. C: The population size will remain constant at the 2010 level. L: Low population expansion scenario. M: Medium population expansion scenario. H: High population expansion scenario. The three rows of panel show the effects of aging and adaptation on the heat-related *YLLs* in 2030s, 2060s, and 2090s, respectively. The heat-related *YLLs* in the future have been subtracted by the heat-related *YLLs* in the 1980s.

**Table 1 ijerph-16-00376-t001:** Mean, range, and specific percentiles for studied variables in Guangzhou during 2010–2015.

Variables	Mean	Min	25th	75th	Max
Total *YLLs* per day	2408.7	982.7	2035.2	2716.5	4171.6
Gender					
Males	1455.1	522.5	1208.8	1666.4	2652.7
Females	953.5	360.2	785.2	1095.1	1982.9
Age groups					
<65 years	1486.6	588.8	1206.9	1712.0	2987.0
≥65 years	922.1	355.0	766.4	1066.1	1731.8
Mean temperature (°C)	21.9	4.8	17.3	27.2	32.2
Wind speed (m/s)	2.3	0.3	1.5	2.7	9.5
Relative humidity (%)	77.6	30.0	71.0	85.0	100.0
Mean temperature (°C) during 1980s	22.0	3.9	16.8	27.3	32.7
SO_2_ (μg/m^3^)	22.2	2.0	13.1	27.7	106.5
NO_2_ (μg/m^3^)	44.9	9.8	29.5	54.5	345.8
PM_10_ (μg/m^3^)	68.9	9.6	44.1	87.0	419.8

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
