# Peer review of "Modification Effects of Population Expansion, Ageing, and Adaptation on Heat-Related Mortality Risks Under Different Climate Change Scenarios in Guangzhou, China"

_ijerph, 2019, doi:10.3390/ijerph16030376_

Round 1
Reviewer 1 Report
In general, this manuscript looks at a topic that interests a wide audience. The analysis is sound and complete, and the framing with 'years of life lost' is particularly interesting. However, I do have concerns about the clarity and flow of this manuscript. I suspect a reader less familiar climate projections or heat-health would find this manuscript very confusing in its current form. There are choices that should be better justified, 'urban expansion' should be rephrased, and the wealth of information in the Supplementary document should be referred to throughout the manuscript. The following are my specific comments:
line 70: Why study Guangzhou? I think the authors need to give a better justification than saying it's "the third largest metropolis in China".
line 75: Need a citation for DLNMs and refer readers to the Supplementary Information.
line 76: The authors should cite IPCC AR5 for the RCPs, and refer the reader to the Supplementary Information for these scenarios.
line 129: why these 27 GCMs? How many ensemble members (or realisations) are used from each model?
line 134: The three bias-correction methods need to be cited.
It is unclear to me how bias correction is done. I think there needs to be a section in the supplementary information about this. My biggest concern is some of the CMIP5 output have very coarse horizontal grid (e.g. 2.8 x 2.8 degree in CanESM2), and downscaling to 0.5 x 0.5 would introduce uncertainty.
line 143 and thereafter: I strongly suggest the authors replace 'urban expansion' with 'population expansion'. Urban expansion sounds to me you take changes in land use due to urbanisation into account (which will affect temperatures), when in fact you only consider population changes.
line 155: I think the justification of 8.92% and 4.6% adaption changes need to be made here. I notice relevant information is in Discussion but that should be moved here. Also justification of S3 for 0.2 C MMT increase per decade is in Supplementary Information, which should be mentioned here.
line 157: Need to refer to the definition of MMT by adding '(See Supplementary Information)'.
line 162: I am concerned about the 0-1 day lag in the DLNMs. There are clearly heat effects on YLL up to ~ 2.5 days. Why didn't the authors choose a 0-2 days lag instead?
line 191: "low and medium scenarios' is very confusing here, given the previous sentence is about low/middle/high emission scenarios. I think the authors need to distinguish between low/middle/high emission scenarios from low/medium/high population expansion scenarios throughout the manuscript, if that's what they mean.
line 191: The authors also need to define what low, medium and high population expansion scenarios are in the main manuscript. These are mentioned in the caption of Table S2 but Table S2 is not mentioned in main text. I suggest defining the scenarios and add "(please refer to Table S2 for details)" in this sentence.
From Table S2, it looks to me low population expansion scenario = 10th percentile of future population projections, medium = 50th percentile and high = 90th percentile. If this is correct, then I suggest showing the 80% projection interval (i.e. 10th to 90th percentiles) in Figure S3 (dashed lines).
line 219-220: "constant urban expansion scenario" probably means "the scenario in which population remains constant at the 2010 levels" here. If so, it should be rephrased to the latter, as "constant urban expansion scenario" sounds like you have a scenario where urbanisation happens at a constant rate.
line 227: I think the reason for the 2090s YLLs lower in the low and medium population expansion scenarios than under constant population should be given here, not in the discussion.
line 260: What is adaptation S3? Figure 4 says "adaptation to high temp. increase by 0.1 C per decade", Table S4 says "adaptation to high temp. increase by 0.2 C per decade", whereas line 231 says "0.2 C MMT increase per decade". At least one of these are inaccurate.
line 278: I would replace "reverse" with "reduce".
line 368: Is there a chance that the reduction in cold related YLLs outweighs the increase in heat-related YLLs?
Acknowledgement: Official acknowledgement of the CMIP5 output (https://cmip.llnl.gov/cmip5/citation.html) should be used.
Author Response
In general, this manuscript looks at a topic that interests a wide audience. The analysis is sound and complete, and the framing with 'years of life lost' is particularly interesting. However, I do have concerns about the clarity and flow of this manuscript. I suspect a reader less familiar climate projections or heat-health would find this manuscript very confusing in its current form. There are choices that should be better justified, 'urban expansion' should be rephrased, and the wealth of information in the Supplementary document should be referred to throughout the manuscript. The following are my specific comments:
line 70: Why study Guangzhou? I think the authors need to give a better justification than saying it's "the third largest metropolis in China".
Response: Thanks for your suggestion, and we have added more information to justify the study setting selection (Line 130-135 in Page 3).
line 75: Need a citation for DLNMs and refer readers to the Supplementary Information.
Response: We have added more detailed information in the methods section (Line 236 in Page 6).
line 76: The authors should cite IPCC AR5 for the RCPs, and refer the reader to the Supplementary Information for these scenarios.
Response: Revised accordingly (Line 159 in Page 4).
line 129: why these 27 GCMs? How many ensemble members (or realisations) are used from each model?
Response: We collected all GCMs from the CMIP5. However, data in some GCMs was seriously missing for some RCP scenarios. After excluding those GCMs with missing data, only 27 GCMs were finally included in this study. For each GCM model, we used a single ensemble because we aimed to solve the uncertainty associated with alternative model formulations form different modelling groups (Line 159-160 in Page 4).
line 134: The three bias-correction methods need to be cited.
It is unclear to me how bias correction is done. I think there needs to be a section in the supplementary information about this. My biggest concern is some of the CMIP5 output have very coarse horizontal grid (e.g. 2.8 x 2.8 degree in CanESM2), and downscaling to 0.5 x 0.5 would introduce uncertainty.
Response: We have added the citation for the three bias-correction methods (Line 164 in Page 4), and have added more detailed information on the methods of temperate bias correction in the supplementary materials. In order to reduce the uncertainty of downscaling, we critically assessed the downscaling method. The results showed good performance of downscaling method (See Figure S2).
line 143 and thereafter: I strongly suggest the authors replace 'urban expansion' with 'population expansion'. Urban expansion sounds to me you take changes in land use due to urbanisation into account (which will affect temperatures), when in fact you only consider population changes.
Response: Revised accordingly.
line 155: I think the justification of 8.92% and 4.6% adaption changes need to be made here. I notice relevant information is in Discussion but that should be moved here. Also justification of S3 for 0.2 C MMT increase per decade is in Supplementary Information, which should be mentioned here.
Response: Revised accordingly (Line 205-221 in Page 5).
line 157: Need to refer to the definition of MMT by adding '(See Supplementary Information)'.
Response: We have given the detailed definition of MMT in the text (Line 222-223 in Page 5).
line 162: I am concerned about the 0-1 day lag in the DLNMs. There are clearly heat effects on YLL up to ~ 2.5 days. Why didn't the authors choose a 0-2 days lag instead?
Response: We selected the lag structure based on the significance of temperature’s effects. On the lag 2, the effect of temperature was not statistically significant (Line 245-246 in Page 6, and Figure S4).
line 191: "low and medium scenarios' is very confusing here, given the previous sentence is about low/middle/high emission scenarios. I think the authors need to distinguish between low/middle/high emission scenarios from low/medium/high population expansion scenarios throughout the manuscript, if that's what they mean.
Response: In order to avoid the confusion between emission scenarios and population expansion scenarios, we have deleted the “low/middle/high” words in the text (Line 345-346 in Page 9).
line 191: The authors also need to define what low, medium and high population expansion scenarios are in the main manuscript. These are mentioned in the caption of Table S2 but Table S2 is not mentioned in main text. I suggest defining the scenarios and add "(please refer to Table S2 for details)" in this sentence.
Response: We have added more detailed information on the three scenarios in the methods section (Line 189-194 in Page 5).
From Table S2, it looks to me low population expansion scenario = 10th percentile of future population projections, medium = 50th percentile and high = 90th percentile. If this is correct, then I suggest showing the 80% projection interval (i.e. 10th to 90th percentiles) in Figure S3 (dashed lines).
Response: We have added the 80% projection interval in Figure S3.
line 219-220: "constant urban expansion scenario" probably means "the scenario in which population remains constant at the 2010 levels" here. If so, it should be rephrased to the latter, as "constant urban expansion scenario" sounds like you have a scenario where urbanisation happens at a constant rate.
Response: Thanks for your suggestion and we have revised it to “comparing with the scenario of constant population size at the 2010 level” (Line 377-378 in Page 11).
line 227: I think the reason for the 2090s YLLs lower in the low and medium population expansion scenarios than under constant population should be given here, not in the discussion.
Response: Revised accordingly (Line 388-389 in Page 11).
line 260: What is adaptation S3? Figure 4 says "adaptation to high temp. increase by 0.1 C per decade", Table S4 says "adaptation to high temp. increase by 0.2 C per decade", whereas line 231 says "0.2 C MMT increase per decade". At least one of these are inaccurate.
Response: Sorry for our mistakes, and the adaptation S3 means that people’s adaptation to high temperature will increase by 0.2°C per decade, which is indicated that the MMT increase 0.2°C per decade. We have corrected the mistakes in the Figure 4 (Page 13).
line 278: I would replace "reverse" with "reduce".
Response: Reverse means turn back. We use “reverse” to describe that “if we cannot reverse (turn back) the trend of global warming in the near future, …”.
line 368: Is there a chance that the reduction in cold related YLLs outweighs the increase in heat-related YLLs?
Response: In this study, we did not assess the cold related YLLs. Hence, we can not estimate how much of heat-related YLLs could be offset by the cold related YLLs. However, previous several studies suggested that the reduction in cold related mortality could largely offset and even outweigh the increase in heat-related mortality (Martin et al., 2011; Vardoulakis et al., 2015).
Acknowledgement: Official acknowledgement of the CMIP5 output (https://cmip.llnl.gov/cmip5/citation.html) should be used.
Response: Revised accordingly (Line 590-595 in Page 19).
Reviewer 2 Report
Paper is not very clear , with poor description of methods and rationale doesn't seem to be well thought through.
Some important issues :
- time series study on a limited time period 2010-2015, iclusive on confounders (weetahe variables and air pollution) which are not accounted for in the future scenrios, so what is the point of including them?
- main issue is that authors model tempertaure-mortality association on 2010-2015 and then use this estimate to define a baseline condition in 1980. The temporal varaition in heat-realted mortality has been described abundantly in the literature and this is un acceptable especially as you describe the already existing temperature increase and population change. abseline year shoudl refer to a year comprised in the 2010-2015 on which you construct the effect estimates. The current estimates may have some serious misclassifictaion and bias due to this aspect.
- Methods not clearly illustrated. how were YLL calculated? How were future estimates calculated? How are adaptation scenarios is France, New York and Shanghai comparable to the study area? why were these selected? popluation characteristics are very different (high proportion of lederly in US and France) adaptation measures have been in place for over 10 years now and more recent papers a reduction in heat related effects. The change in effects (reduction) implies that some adaptation is introduced, is there currently something being done at local level to suggest these effects are plausible? If not the studies mentioned are not comparable. Why not use WHO or multi city adaptation scenarios? or at least include more as sensitivity analysis.
- sensitivity nalysis should not be on DLNM model bu ton the future impact estimates, using different baseline period, error estimates of RCP scenarios, different future periods of comparison.
-
Author Response
Paper is not very clear , with poor description of methods and rationale doesn't seem to be well thought through.
Response: We have added more detailed information about the methods in the manuscript, including data collection, future temperature projection, heat effects estimation, and statistical analysis.
Some important issues :
- time series study on a limited time period 2010-2015, iclusive on confounders (weetahe variables and air pollution) which are not accounted for in the future scenrios, so what is the point of including them?
Response: It has been demonstrated that air pollutants and other weather variables besides temperature can also affect mortality risks. Therefore, we need to adjust for these variables to exclude their confounding effects. Otherwise, the effects of temperature on mortality and further YLLs could be biased.
- main issue is that authors model tempertaure-mortality association on 2010-2015 and then use this estimate to define a baseline condition in 1980. The temporal varaition in heat-realted mortality has been described abundantly in the literature and this is un acceptable especially as you describe the already existing temperature increase and population change. abseline year shoudl refer to a year comprised in the 2010-2015 on which you construct the effect estimates. The current estimates may have some serious misclassifictaion and bias due to this aspect.
Response: Thanks very much for your comments. The selection of baseline was controversial in previous studies investigating health effects of future climate change. However, many studies employed 1980s as baseline because this decade is at the centre of the conventional climatological baseline period of 1970-1999 (Li et al., 2013).
- Methods not clearly illustrated. how were YLL calculated? How were future estimates calculated? How are adaptation scenarios is France, New York and Shanghai comparable to the study area? why were these selected? popluation characteristics are very different (high proportion of lederly in US and France) adaptation measures have been in place for over 10 years now and more recent papers a reduction in heat related effects. The change in effects (reduction) implies that some adaptation is introduced, is there currently something being done at local level to suggest these effects are plausible? If not the studies mentioned are not comparable. Why not use WHO or multi city adaptation scenarios? or at least include more as sensitivity analysis.
Response: Thanks very much for your comments. We have added more detailed information about the methods in the manuscript, including the calculation process of YLL. Adaptation estimation in the future is a crucial issue in the health risk assessment of climate change. In previous studies, methods used to account for adaptation included analogue cities, analogue summers and assuming adaptation to heat for a pre-determined number of degree Celsius. However, such methods usually lead to uncertainties to the adaptation estimation. For example, it is usually difficult to quantitatively assess the adaptation level of analogue cities. In recent years, several studies used the past declines in vulnerability to temperature to define the change of adaptation for a city. Ideally, we would have assessed the change of adaptation in Guangzhou using a long period meteorological and health data. However, we don’t have such data. The only study that assessed the long-term change of temperature on mortality in China was conducted in Shanghai, but this study included only 32 years’ data. Although the socioeconomic characteristics in Guangzhou and Shanghai are similar, the 30 years was not long enough to totally capture the long-term variation of people’s adaptation. In addition, Yang et al. assessed only heat effects on cardiovascular mortality. Therefore, we included other two scenarios from New York City and France. We believe that these estimates of heat effects derived from a broad range of adaptation scenarios provides uncertainty information for policy makers concerned with adaptation planning for climate change and health. On the other hand, these adaptation scenarios are also sensitivity analyses. All three adaptation scenarios have illustrated that adaptation could largely offset the heat effects of climate change in Guangzhou. We have given adequate discussion on this issue in the discussion section (Line 508-527 in Page 17).
- sensitivity nalysis should not be on DLNM model bu ton the future impact estimates, using different baseline period, error estimates of RCP scenarios, different future periods of comparison.
Response: Thanks for your suggestions, and we have deleted the sensitivity analysis on the DLNM model (Line 414-417 in Page 12). Actually, we included many scenarios in the results, including three RCP scenarios, 27 GCMs, three population increase scenarios, and three adaptation scenarios. All these scenarios could provide uncertainty information for policy makers. On the other hand, the results of these scenarios were also sensitivity analyses, which have demonstrated the robustness of our findings.
Reviewer 3 Report
The article will be valuable as an attempt of projecting future change of heat-related mortality risk under global and urban warming. However, the manuscript leaves much to be desired in presentation, so I would recommend major revision. In particular, the detailed procedure of analysis should be written in the main text, not in the Supplement.
[Main comments]
@ The Supplement is so frequently referenced from the main text, and a large part of the procedure of analysis is written in the Supplement. Moreover, there is substantial overlapping in the main text and the Supplement. I think reference to the Supplement from the main text should be avoided as far as possible, and the procedure of analysis, written from Page 4 to Page 6 of the Supplement, should be written in the main text.
@ I would appreciate the usage of more than one adaptation scenario, but the three scenarios are all based on a fixed rate in time (8.92%, 4.6%, and 0.2C per decade). In reality, the degree of adaptation is likely to depend on the climate scenario; a faster warming will be accompanied by a higher degree of adaptation. It is recommended to make some comments on the possible dependence of adaptation on scenarios and its effect on the results in the Discussion.
@ Please write the exact meaning of "years of life lost (YLL)" in the present study. For example, what is the reference age for YLL?
@ The threshold temperature is different between the analysis for the total population and that for each gender and age group. Doesn't it cause inconsistency between the results for the total population and each gender/age group? It may be better to add some explanation and/or discussion.
@ A more detailed explanation is needed for Fig.S4. How was the "lag effect" defined and calculated?
[Other comments]
@ It is better to add a geographical name (Guangzhou, China) to the Title.
@ Section 1.2 and the first paragraph of 1.3 are not suitable for the Introduction. This kind of thing should be written in the Discussion. The sentences "We searched PUBMED, --- on heat-related health." from Line 61 to 66 are also unnecessary.
@ Line 201 "All the effects of temperature on YLLs were adjusted ---" --- Please explain the meaning of "adjustment".
@ Please add an explanation to the three rows in Figs.4 and 5, although they appear to correspond to 2030s, 2060s, and 2090s.
@ Fig.4 has some negative values of YLL. Do they indicate deviation from the present state (or 1980s)? Please check, and add explanation.
Author Response
The article will be valuable as an attempt of projecting future change of heat-related mortality risk under global and urban warming. However, the manuscript leaves much to be desired in presentation, so I would recommend major revision. In particular, the detailed procedure of analysis should be written in the main text, not in the Supplement.
Response: Thanks for your positive comments, and we have added more detailed information about the methods in the manuscript.
[Main comments]
@ The Supplement is so frequently referenced from the main text, and a large part of the procedure of analysis is written in the Supplement. Moreover, there is substantial overlapping in the main text and the Supplement. I think reference to the Supplement from the main text should be avoided as far as possible, and the procedure of analysis, written from Page 4 to Page 6 of the Supplement, should be written in the main text.
Response: Thanks for your comments, and we have added more detailed information about the methods in the manuscript.
@ I would appreciate the usage of more than one adaptation scenario, but the three scenarios are all based on a fixed rate in time (8.92%, 4.6%, and 0.2C per decade). In reality, the degree of adaptation is likely to depend on the climate scenario; a faster warming will be accompanied by a higher degree of adaptation. It is recommended to make some comments on the possible dependence of adaptation on scenarios and its effect on the results in the Discussion.
Response: Revised accordingly (Line 512-524 in Page 17).
@ Please write the exact meaning of "years of life lost (YLL)" in the present study. For example, what is the reference age for YLL?
Response: We have added the exact meaning of YLL in the methods section. YLLs were calculated by matching each death by age and sex to the life table of China for the year 2010. Total daily YLLs were calculated by summing the YLLs for all deaths on the same day (Line 140-141 in Page 4).
@ The threshold temperature is different between the analysis for the total population and that for each gender and age group. Doesn't it cause inconsistency between the results for the total population and each gender/age group? It may be better to add some explanation and/or discussion.
Response: Revised accordingly (Line 541-544 in Page 18).
@ A more detailed explanation is needed for Fig.S4. How was the "lag effect" defined and calculated?
Response: Revised accordingly.
[Other comments]
@ It is better to add a geographical name (Guangzhou, China) to the Title.
Response: Revised accordingly.
@ Section 1.2 and the first paragraph of 1.3 are not suitable for the Introduction. This kind of thing should be written in the Discussion. The sentences "We searched PUBMED, --- on heat-related health." from Line 61 to 66 are also unnecessary.
Response: We have deleted the first three paragraphs from the introduction (Line 68-100 in Page 2-3).
@ Line 201 "All the effects of temperature on YLLs were adjusted ---" --- Please explain the meaning of "adjustment".
Response: The effects of temperature on YLLs may be confounded by other factors, including the long-term trend of mortality, wind speed, day of week. In the DLNM model, we adjusted for these potential confounders.
@ Please add an explanation to the three rows in Figs.4 and 5, although they appear to correspond to 2030s, 2060s, and 2090s.
Response: Revised accordingly (see Figs.4 and 5).
@ Fig.4 has some negative values of YLL. Do they indicate deviation from the present state (or 1980s)? Please check, and add explanation.
Response: Taking the lower-left panel in Fig.4 for example, the negative values of YLLs indicate that if the population expansion follows the low scenario, the heat-related YLLs in the total population in the 2090s may be lower than that in the 1980s.
Reviewer 4 Report
This manuscript is very interesting and has a creative methodological scope in my opinion. There is no doubt that the material is relevant and it works with a strategical topic nowadays (adaptation). This manuscript has strategical relevance to China due to China's classical demographic issue as well.
I would like to tell about the figures 4 and 5 (that is a suggestion). They are very interesting and they are the core of the paper results, anyway, I think that they could be more suitable for comparisons if the axis Y have the same scale range within the three temporal scenarios (2030s, 2060s and 2090s). This technical change could permit a better analysis between 2030s, 2060s and 2090s periods. Nevertheless, the pattern that is being used valorizes the levels of adaptation within a specific period.
Author Response
This manuscript is very interesting and has a creative methodological scope in my opinion. There is no doubt that the material is relevant and it works with a strategical topic nowadays (adaptation). This manuscript has strategical relevance to China due to China's classical demographic issue as well.
Response: Thanks very much for your positive comments.
I would like to tell about the figures 4 and 5 (that is a suggestion). They are very interesting and they are the core of the paper results, anyway, I think that they could be more suitable for comparisons if the axis Y have the same scale range within the three temporal scenarios (2030s, 2060s and 2090s). This technical change could permit a better analysis between 2030s, 2060s and 2090s periods. Nevertheless, the pattern that is being used valorizes the levels of adaptation within a specific period.
Response: Thanks very much for your suggestion. In the primary version, we have tried the figure with same Y scale. However, the boxes in some panels were too small to clearly be shown, because the metrics of Y scale between different periods largely varied.
Round 2
Reviewer 3 Report
I appreciate the authors' effort of revision. However, I am still confused about the meaning of "YLL" in the article. The authors state "The YLLs estimated during the baseline period were subtracted from heat-related YLLs for future periods." in Lines 101 to 103 on Page 3. This means that relative values of YLL are used, as done in Fig.4. However, it appears that absolute values of YLL are also used in some parts, as in Fig.2. If I understand correctly, it is recommended to take some means to avoid confusion as to the meaning of "YLL" in the text and figures. If absolute and relative values of YLL are both used, then it will be better to use different denotations.
Author Response
Response: Thanks very much for your further comments. In many previous studies that have assessed the health impacts of future climate change, they have examined the temperature related excess deaths or mortality risks. This method of giving the equal weight to every death occurring at very different ages may distort policy priorities and decision making. Years of life lost (YLL) is an indicator of premature mortality that accounts for the age at which deaths occurred by giving greater weight to deaths at younger ages. This indicator of YLL could provide more information for quantifying premature mortality compared to mortality risk, and hence is worth being widely used in future studies.
In Figure 2, we have to first estimate the exposure-response relationships between temperature and YLLs, and the attributed YLL of each temperature was estimated, which was used in the estimations of heat-related YLLs in different periods. Therefore, we used the absolute values of YLL in Figure 2.
In the studies of health impacts of climate change, we need to usually estimate the health impacts of temperature changes during a long-term period. Hence, the results can be attributed to the change of climate. In that case, we need to select a time period as the baseline (1980s). All the heat-related YLLs in the future time periods must be subtracted by the heat-related YLLs during the baseline period. We have used different denotations in the manuscript (Figure 3 in Page 9, Figure 4 in Page 12, Figure 5 in Page 13, and supplementary materials).